# Source Apportionment of PM10 as a Tool for Environmental Sustainability in Three School Districts of Lecce (Apulia)

**Tiziana Siciliano** [1,*], **Antonella De Donno** [2], **Francesca Serio** [3] and **Alessandra Genga** [3]

1 Department of Mathematics and Physics "Ennio De Giorgi", University of Salento, 73100 Lecce, Italy
2 Department of Experimental Medicine, University of Salento, 73100 Lecce, Italy;
  antonella.dedonno@unisalento.it
3 Department of Biological and Environmental Sciences and Technologies, University of Salento,
  73100 Lecce, Italy; francesca.serio@unisalento.it (F.S.); alessandra.genga@unisalento.it (A.G.)
* Correspondence: tiziana.siciliano@unisalento.it; Tel.: +39-0832-297067

**Abstract:** Air pollution is a great threat to the sustainable development of the world; therefore, the improvement of air quality through the identification and apportionment of emission sources is a significant tool to reach sustainability. Single particle analysis, by means of a scanning electron microscope equipped with X-ray energy dispersive analysis (SEM-EDS), was used to identify the morphological and chemical properties of the PM10 particles in order to identify and quantify the main emission sources in three areas of Lecce, a city in the Apulia region of southern Italy. This type of characterization has not yet been performed for the Lecce site, but it is of particular importance to identify, based on the shape of the particles, the natural sources from the anthropogenic sources that are responsible for the serious health effects of the inhabitants. Three primary schools located in peripheral areas of the city were chosen for the sampling: "School 1" (A site), "School 2" (B site), and "School 3" (C site) to carry out a study of the air quality. The A site is characterized by a greater presence of calcium sulphates probably due both to construction activities present during sampling and to reactions between Ca particles and the sulfur present in the atmosphere. At the C site, there is a relative numerical abundance of different groups of particles that present, in the EDS spectrum, an enrichment in sulfur. At the B site, the number of particle groups is intermediate compared to that of the other two sites. With the source apportionment technique, ten emission sources were identified: combustion, soot, industry, soil, carbonates, sea salt, calcium sulfates, SIA, biological particles, and others. In PM10, the three sites are more affected by the soil source, with an effect greater than 60%.

**Keywords:** PM10; SEM-EDS; source apportionment; emission sources; environmental sustainability

## 1. Introduction

In recent decades, the European Member States have focused their efforts on sustainability through the safeguard and protection of the environment, the improvement of the well-being of people and the planet, health defense, and sustainable economic growth. Strategic initiatives have been undertaken for a green transition. One of these is the reduction of air pollution, which has man's daily activities as its primary source. The study of atmospheric pollutants and the identification and apportionment of emission sources are, therefore, good tools for environmental policy regarding atmospheric pollution reduction. The importance of improving air quality is related to the threat to the health of humans as well as all other forms of life from exposure to air pollution. Multiple research investigations have demonstrated a connection between outdoor air pollution and the development of a number of cardiovascular and respiratory diseases, including chronic obstructive pulmonary disease (COPD), asthma, some types of cancer [1–4], the transmission of infectious diseases [5], immune dysfunction, endocrine and neurodegenerative disorders, and fetal complications [6].

Children are especially vulnerable to the effects of air pollution due to their smaller size, quicker rate of growth, and relatively undeveloped organs, as well as other body processes, including the immune system and cell repair mechanisms. In addition, children are more vulnerable to the impacts of pollution because they breathe more quickly than adults, absorbing more pollutants, and because they are located closer to the ground, where some pollutants have the highest concentrations. According to the World Health Organization, young children are a high-risk category for both the short- and long-term health effects of air pollution. They are also more sensitive to indoor air pollution in households where polluting fuels and technologies are frequently used for cooking, heating, and lighting [7]. The evidence showing adverse effects on children's health has grown significantly since the American Academy of Pediatrics policy statement on ambient air pollution was published in 2004 [8].

The PM investigation in the city of Lecce has been carried out in several studies [9–12], where primary and secondary contributions to PM by means of bulk chemical analyses were carried out. Bulk chemical analyses provide averaged information on the composition of the PM, and it is possible to trace the emission sources through statistical approaches by considering the correlations between the analytes present in the sample. Single particle analysis allows us to obtain non-mediated but detailed information on each particle present on the sampled membrane, determining the morphological parameters (i.e., shape, size, aspect ratio, etc.) and chemical parameters of each analyzed particle. The power of this technique is to have more detailed information on the sources and the possibility to identify sources with the same chemical composition but different morphology; moreover, it is possible to distinguish among sources characterized by the same elements in different ratios. Furthermore, single particle analysis is able to identify and quantify a very large number of sources by just analyzing one sample; instead, to have the same information using bulk analyses, a very large number of samples is generally necessary. Single particle analysis has been more commonly used in the last few years for the study of PM [13,14], but a thorough morpho-chemical characterization of the particles is, in our opinion, lacking in Lecce.

The aim of this research was to study the air quality in the city of Lecce in order to evaluate the presence of chemicals belonging to anthropogenic sources that could have a negative effect on children's health. Children spend a great part of their lives in school, and in this work, we studied the main sources present in school districts. For this reason, PM10 samples were collected and analyzed by scanning electron microscopy coupled with X-ray energy dispersive analysis (SEM-EDS) in three school districts in order to qualitatively/quantitatively analyze the sources that impact the sites, identify groups of particles based on their size, shape, and chemical composition, study the size distribution of particles belonging to the same sources, and compare the sources among the studied sites using source apportionment methods.

## 2. Materials and Methods

### 2.1. Site Description—Sample Collection

The sampling locations are situated in Lecce (Figure 1), a city in Southeast Italy, on the Salento peninsula in the Apulia region ($40°23'00''$ N, $18°11'00''$ E).

The Salento peninsula is flat with minor elevations above sea level; in fact, it is distinguished by little ridges, the so-called "Serre salentine" [15]. "Mediterranean" climate can be used to describe the Salento peninsula's weather. The two main winds are quite hot: the Scirocco, which blows from the southeast and is very humid because it originates from the Eastern Mediterranean Sea basin, and the Libeccio, which blows from the southwest and is drier because it releases its humidity on the mountains of Sicily and Calabria before it reaches the western shore of Salento. NNW wind occurs more frequently during the summer [16]. With a mean value of 790 mm/year in the eastern section and only 590 mm/year in the western part, the yearly rainfall is not uniform. But because of a sharp decline in average rainfall, there has been a protracted drought. As a result, the Salento

peninsula's climate can be characterized as virtually semi-arid (very high temperatures and insufficient precipitation): the dryness tends to occur in summer, while the precipitation is concentrated in autumn and winter, with only very few occurrences at the end of summer [17].

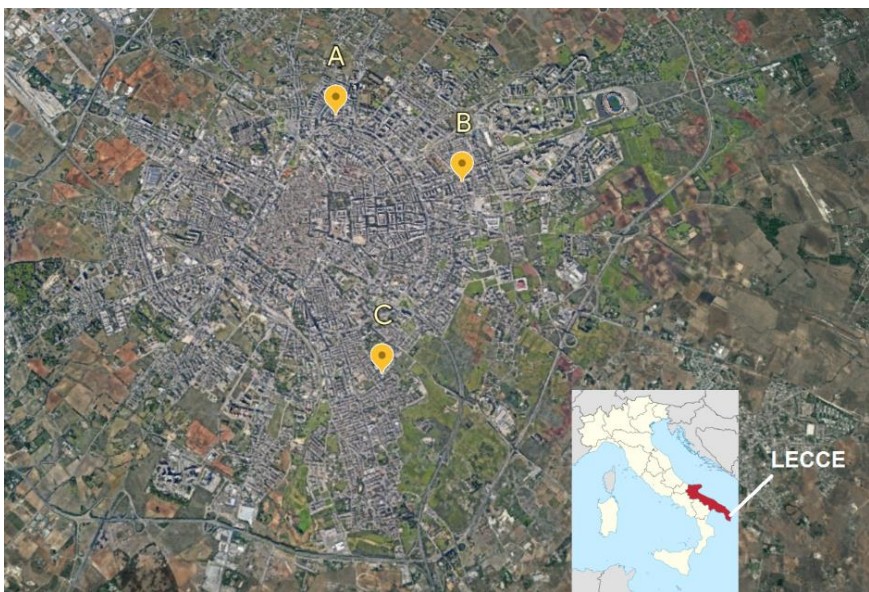

**Figure 1.** Map showing the location of the three measurement sites in Lecce town. Map provided by Google Earth.

In the peripheral area of the city of Lecce, three measuring sites were chosen that shared comparable qualities (Figure 1). To conduct research on the air quality for the children who attend these schools, three primary schools—"School 1" (A site), "School 2" (B site), and "School 3" (C site)—located in various parts of the city with varying traffic intensities have been identified. The PM10 samples were taken at three locations between May and June 2015, namely on 4 June 2015 at the A site, 29 May 2015 at the B site, and 12 May 2015 at the C site, from 10:30 a.m. to 4:30 p.m. The ARPA Puglia Agency contributed the meteorological data for the sampling days, which are depicted in Figure 2 for the city of Lecce. According to Figure 2, the wind primarily blows from the north at the A and B sites and from the north-northeast sector at the C site, with average speeds of 2.7 ms$^{-1}$ and 3 ms$^{-1}$, respectively, during the sample hours. The three days that correspond to the sample time slot under consideration saw an average temperature of 23 °C at the C site, 21 °C at the B site, and 28 °C at the A site. Additionally, a backward trajectory at 120 h was estimated to identify the source and route of the air masses on a broad scale (see bottom of Figure 2). Using the NOA Hysplit Model and GDAS (Global Data Assimilation System) Meteorological Data, the trajectories were generated for the three sites on their respective sample days, with the endpoint at 12:00 UTC and at altitudes of 500 (representative of lower altitude transport), 1000, and 2000 m (representative of the uppermost part of the boundary layer and the mean transport winds at the synoptic scale) AGL (above ground level). The back trajectories show different origins and have traveled different distances in the same time range. More specifically, trajectories reaching site A have a greater national influence as they spend most of their time in central Italy (red line) and surrounding Eastern European states (blue and green lines). The air masses reaching site B cover a larger geographical area, starting from the Atlantic Ocean and passing over several regions of Northern and Eastern Europe, including Scotland, Germany, Croatia, and Bosnia-Herzegovina. Air masses originating in Northwestern Europe (France and Germany) arrive at site C and, traveling through Poland and Romania, reach the regions of southern Europe. Therefore, the air masses at the three sites could have substantially different compositional characteristics.

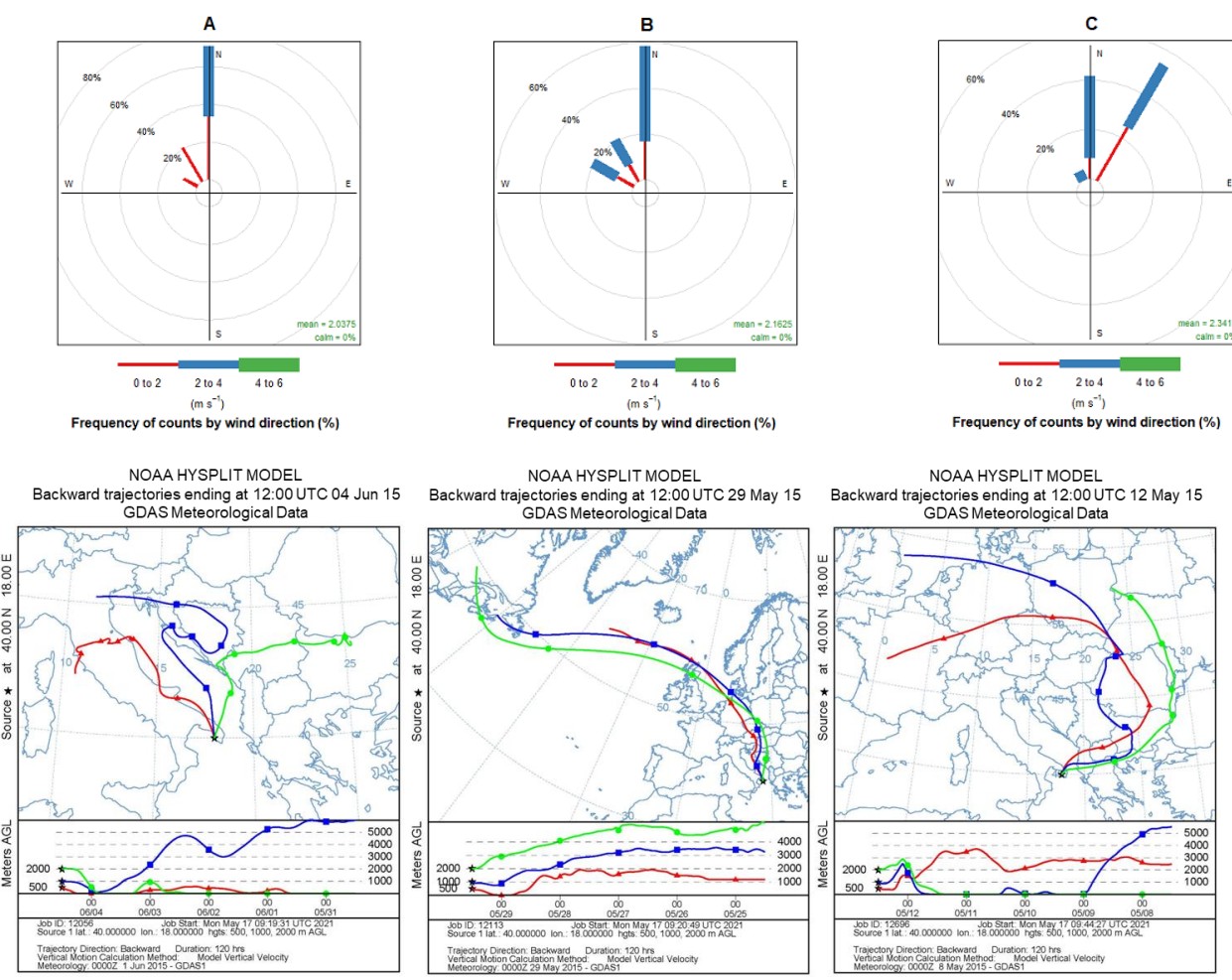

**Figure 2.** Rose wind (**top**) and backward trajectories (**bottom**) relating to sampling days for measurement sites A, B and C (4 June 2015, 29 May 2015, 12 May 2015, respectively). In subfigures the model set up is reported.

### 2.2. Subsection Site Description—Sample Collection Scanning Electron Microscopy (SEM)

All samples for electron microscopy were collected on polycarbonate membranes (47 mm diameter, 0.8 μm pore size, Nuclepore Whatman) using a low volume sampler (Tecora Charlie HV) operating at a flow rate of 2.3 m³h⁻¹. The sampling time was set at 6 h to avoid an overload of the sample carrier, as this would prevent a reliable analysis of individual particles in the SEM.

Particles collected on a polycarbonate filter were coated with a thin gold film (10 nm in thickness) to obtain the samples that were electrically conductive. In addition, the particles were examined with a scanning electron microscope (Tescan Vega/LMU, Libušina, Kohoutovice, Czech Republic) in order to assess size and morphology. Elemental composition of particles was investigated with an energy-dispersive microanalysis system (EDS, Quantax Bruker (Bruker, Billerica, MA, USA) with a lower detection limit per element of less than 0.1%) equipped with the microscope. SEM images from each sample were taken at magnifications of 10,000×, 5000×, and 2500×, and the particles in the size range of 0.4 to 1 μm, of 1 to 2.5 μm, and of 2.5 to 10 μm were analyzed, respectively. The weight percentages of 23 elements (C, N, O, F, Na, Mg, Al, Si, P, S, Cl, K, Ca, Ti, V, Cr, Mn, Fe, Ni, Cu, Zn, Au, and Pb) were calculated by a standardless ZAF correction method in the EDS software (Esprit 1.7). The acquisition times of EDS spectra were typically 20 s to minimize the evaporation of nitrogen-containing compounds, which are often unstable under an electron beam. In this study, EDS analysis is used for particle recognition and not bulk chemical analysis. The single particle study was performed in two steps. In the first

step, the particles were analyzed qualitatively under operator control to define the particle groups. In the second step, quantitative particle analysis was performed automatically to recognize the different particle groups. This procedure is quite time-consuming, leading to a limited number of samples that could be studied.

For the size distribution obtained by individual particle analysis in the SEM, the total number of particles on the sample carrier ($\Delta N$) was estimated for each size range by normalizing the analyzed area to the total area of the deposition spot. The particle number distribution $\frac{dN}{dlogD}$ can be calculated with the following relation:

$$\frac{dN}{dlogD} = \frac{\Delta N}{Qt} \cdot \frac{1}{\Delta logD}$$
$$\Delta logD = logD_2 - logD_1$$

where Q is the flow rate, $t$ is the sampling time, and $D_1$ and $D_2$ are the lower and upper limits of the size interval [18].

## 3. Results and Discussion

### 3.1. Subsection Single Particle Analysis

The particles analyzed in this study had an equivalent spherical diameter (ESD) ranging from 0.4 to 10 μm, which was calculated as $ESD = 2 \cdot \left(\frac{A}{\pi}\right)^{\frac{1}{2}}$ where A is the particle area [19]. About 10,000 particles were analyzed with the SEM/EDS, and, based on morphology and chemical composition, the particles were assigned to eighteen different groups. They were aluminosilicates, silicates, aluminosilicates with sulfur, calcium sulfates, silicate-sulfate-mixed particles, phosphate–sulfate mixed particles, iron oxides, metal oxides, iron mixtures, carbonates, carbonates-silicates, secondary particles, sea salt, aged sea salt, fluorides, soot, biological particles, and remaining carbon-rich particles. All particles, which could not be classified into one of the previous groups, were included in the group labeled "others". This last group mostly consists of agglomerated particles with morphological and compositional characteristics belonging to more than the two groups mentioned above. Based on the results of the single particle analysis carried out with the SEM, the particles, despite being classified according to their chemical composition, also have a different morphology. In fact, particles belonging to the same group can be further distinguished because of their irregular or spherical shape. There is a necessity to isolate and monitor the latter due to their provenance in processes that occur at high temperatures and are therefore of anthropogenic origin. Consequently, the eighteen groups of particles, classified according to the chemical criteria, become twenty-three according to their morphological characteristics.

Below is a brief description of the groups of particles identified in the three sampling sites that contribute significantly to the identification of the emission sources. Further morphological and chemical details of the particle groups obtained with scanning electron microscopy and EDS microanalysis can be found in previous studies [13,14].

Aluminosilicate and silicate particles mainly consist of Al, Si, Na, Mg, Ca, F, and K in their X-ray spectrum (Figure 3b).

The two groups essentially differ in the quantity of Si they contain, and this is reflected in the Al/Si ratio, which is equal to $0.65 \pm 0.26$ for aluminosilicates and $0.14 \pm 0.06$ for silicates. In addition, pure $SiO_2$ particles are also included in the latter group. Aluminosilicates and silicates have both irregular and spherical shapes, as can be seen in Figure 3a. The irregular particles (shape factor equal to $0.69 \pm 0.03$) are of crustal origin and are formed by mechanical processes, while the spherical ones (shape factor equal to $0.92 \pm 0.03$) are of anthropogenic origin and are produced during high-temperature processes such as fossil fuel combustion, coal burning, and metallurgy [20]. In addition to the shape factor, another factor that distinguishes the spherical aluminosilicate particles from the irregular ones is the Si/Al ratio, which is significantly different for the two types of particles. In fact, the Si/Al ratio is equal to $1.3 \pm 0.3$ for particles of industrial origin and $1.83 \pm 0.14$ for those of

crustal origin. These values can be considered in good agreement with those calculated by Cesari et al. [9], even if the methods used to determine this ratio are different. The irregular particles are found in all three investigated sites, i.e., A, B, and C sites; in particular, aluminosilicates represent respectively 8.5%, 17.0%, and 15.5% of the total particle numbers of each site; instead, silicates are found with lower percentages and precisely 5.2%, 7.0%, and 6.5%. The spherical aluminosilicates are more present at the B and C sites, with a percentage around 2%, while the spherical silicates at each site do not exceed 0.5%. The relative abundance of PM10 is depicted in the Supplementary Material (Figure S1).

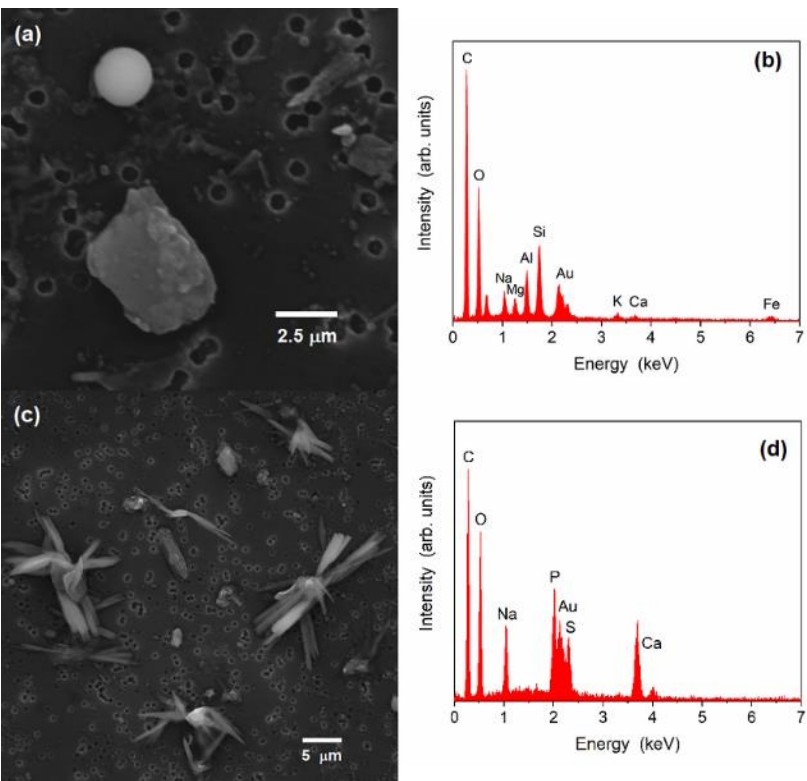

**Figure 3.** (**a**) Typical SEM image of irregular and spherical particles of aluminosilicates; (**b**) relative EDS spectrum; (**c**) typical SEM image of phosphate–sulfate mixed particles; and (**d**) relative EDS spectrum.

With single particle analysis, it was possible to classify two other types of particles that have been labeled as aluminosilicates with sulfur and mixed particles of silicates and sulfates. Aluminosilicates with sulfur show characteristics very similar to aluminosilicates; they differ from the latter due to the presence, in the EDS spectrum, of a greater quantity of sulfur. The silicate–sulfate mixed particles are morphologically irregular and chemically characterized by major elements such as Si, S, and Ca. The quantity of S and Ca does not allow us to include these particles in the previous groups. This combination of elements (gypsum/silicates) can be the result of clusters of such particles or silicates on whose surface secondary sulfur has deposited, or, again, their origin could be due to coal combustion (the coal-fired power plant is about 27 km northwest of the measurement sites) [13,14,21]. At the B and C sites, the percentage number of these particles is about 6%; at the A site, it does not reach 2%.

In this study, another group of particles was identified that were labeled as phosphate–sulfate mixed particles. Morphologically, they appear as sticks of different lengths with a common central part from which these sticks come out (Figure 3c). The EDS spectrum (Figure 3d) reveals the presence of Ca, S, and P as major elements. With our technique, it is difficult to identify their origin, as they could be clusters of Ca phosphates and Ca sulfates or calcium phosphate particles whose presence of sulfur could be of secondary origin. The C site, with

10.4%, had a higher presence of these particles, about double the percentage of the B site; instead, at the A site, these particles slightly exceeded 1%.

Calcium sulfate particles can be easily recognized from the X-ray spectrum (Figure 4d), which is characterized by S and Ca as major elements. As can be seen in Figure 4, they have various morphologies, such as plates, prisms, bars, and needles, as well as particles of irregular shape. Calcium sulfate particles are assigned to natural and anthropogenic sources in equal proportions [22]. Because of their typical crystalline morphologies, they can be identified as gypsum. There are several anthropic processes that can lead to the formation of calcium sulfates [13], and, in particular, particles with elongated morphology might result from aqueous phase formation during secondary atmospheric reactions (i.e., the in-cloud process) [23,24]. Calcium sulfates are the largest group of particles, with 52.6% at the A site, 21% at the B site, and 16% at the C site, and are mainly present in the three size ranges.

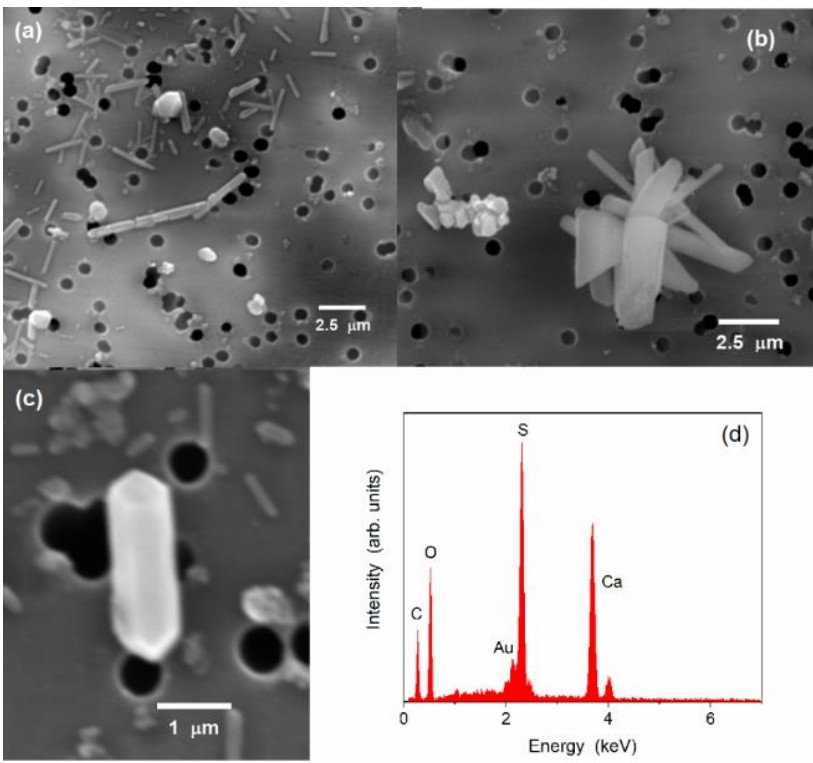

**Figure 4.** Typical SEM images of the calcium sulfates of different shapes: (**a**) small irregular blocks and plates; (**b**) bars; (**c**) prisms; (**d**) typical EDS spectrum of calcium sulfates.

The carbonate group includes carbonates of Ca and Mg and calcium oxides. These particles have an irregular shape, and their X-ray spectrum shows high amounts of C, O, Ca, and Mg. In our study, it was not possible to quantify the carbon element due to the use of a polycarbonate filter as a substrate. This particle group has both natural and anthropogenic origins; it can belong to both the crustal source and anthropogenic activities. In fact, in the proximity of the sampling sites, building construction and demolition work are present. These particles are present in all three sites investigated, and the B site has the highest abundance with 3.3%.

By single particle analysis, it was possible to identify another group of particles labeled as carbonates—silicates due to the presence in the EDS spectrum of a greater quantity of Si and Ca, which makes it chemically different from the groups of silicates and carbonates already exposed. However, this group represents at most 1% of the total particles at each site.

Most of the secondary aerosol particles are mixed compounds of sodium nitrate and/or sulfate and are characterized by different and irregular shapes. These particles also

contain organic material, and SEM analysis and X-ray microanalysis are not sufficient to state whether the organic origin is primary or secondary [18,25,26]. At the three sites, these particles are not equally distributed. In fact, we find a greater number on the C site with 5.5%, followed by the B site with 3.9%, and finally the A site with 0.9%.

Soot particles are found in all three sites investigated with the SEM, mainly resulting from vehicular traffic as well as from incomplete fossil fuel combustion, biofuels, and biomass burning [23,27–30]. As shown in Figure 5, under the microscope, it appears as a branched chain structure consisting of carbonaceous spherical particles with a diameter between 20 and 50 nm. Over time, these structures tend to collapse, forming compact aggregates [13,14]. In the three measurement sites, the percentage abundance of soot is very low; in fact, it is represented by 1.3% at the A site, 1.4% at the B site, and 0.8% at the C site.

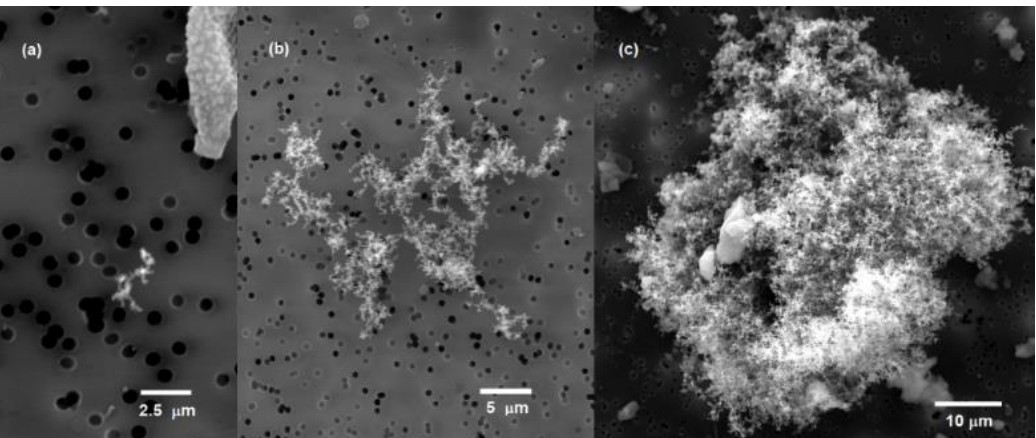

**Figure 5.** Typical SEM images of (**a**) newly formed soot, (**b**) soot branched chain structure, and (**c**) aged aggregate soot.

Biological particles are recognized by their characteristic regular and symmetrical shape and by their presence in the EDS spectrum of elements such as K, P, and S, in addition to C and O. The biological particles mainly include pollen, spores, bacteria, plant fragments, and animal fragments, and in Figure 6, a typical SEM image of them is shown. They are present in all three sites, and their relative abundance is around 2%, with the exception of the A site, where they are less than 1%.

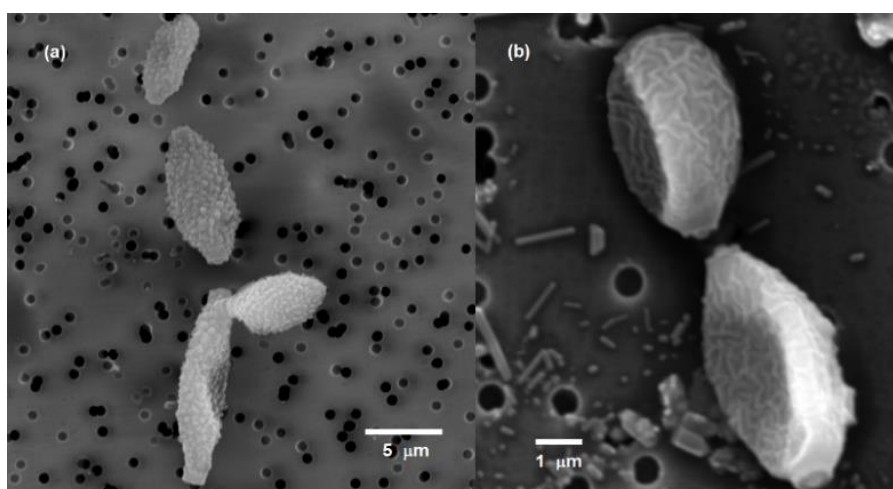

**Figure 6.** (**a**,**b**) typical SEM images of biological particles.

The remaining carbonaceous particles, although chemically made up of C and O as elements, due to their morphology (spherical and irregular shapes), cannot be classified

as soot or as biological particles. In fact, at the three sites, there are isolated spherical particles with an equivalent average diameter of less than 1.37 ± 0.15 µm, the origin of which is of anthropic nature as they are the products of combustion processes. They are very few at the B and C sites, while at the A site they reach 2.4%. Irregular carbonaceous particles are characterized by an aspect ratio greater than 1, which represents the threshold value to define such elongated particles. They are present in the three sites with a relative abundance of around 9–12%.

Very few marine particles have been identified with the SEM, although the city of Lecce is about 10 km away from the Adriatic Sea. It is known from the literature that the chemical composition of marine aerosols depends on the latitude, distance from the coast, and salinity of the sea water. Moreover, the variation in chemical composition is also influenced by meteorological factors such as convection, thermal inversion, air humidity, wind direction, and speed, as well as the occurrence of sea or land breezes [31]. Sea salt particles constitute 0.05% of the total particles found at the B site, and their EDS spectrum consists of major elements such as Na, Ca, and Cl. On the contrary, particles chemically characterized by Na, Ca, and S were found at the A site and constitute 1.89% of the total particles. Aerosol reaching this site from the north was influenced by the maritime environment. In fact, from the back trajectory shown in Figure 2, it can be observed that the air masses, before reaching the site, passed over the sea, loading themselves with sea spray. The replacement of chlorine with sulfur could be explained by the occurrence along the coast of the surface reaction between sea salt and $H_2SO_4$, leading to the formation of aged sea salt [32,33].

With the SEM analysis, particles chemically consisting of Fe and labeled Fe oxides and particles containing other chemical elements such as Al, Si, Ca, Mg, and Na and labeled Fe mixtures were identified. The morphology of these particles discriminates the processes responsible for their formation. The irregularly shaped Fe particles are probably associated with natural sources, especially crustal, and have a relative abundance of 1.01%, 7.38%, and 10.38% at the A, B, and C sites, respectively. The spherical Fe particles (both oxides and mixtures) derive from anthropogenic sources and were found at the B and C sites with a relative percentage abundance of 0.43% and 0.98%, respectively.

In Figure 7, the numerical percentage of the groups of particles as a function of the three size ranges chosen for the electron microscope observations is reported. This analysis was performed for each of the three measurement sites.

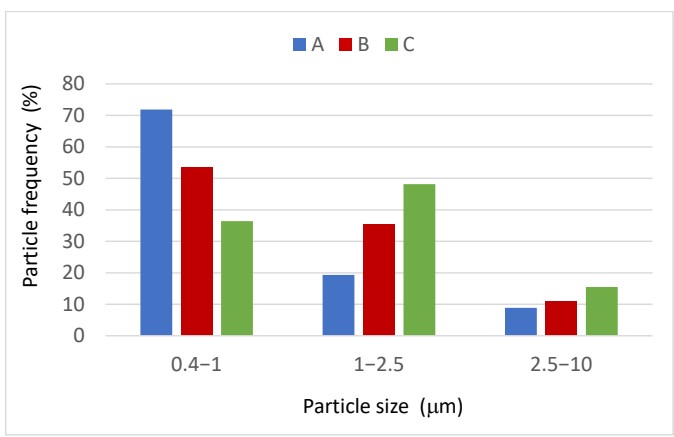

**Figure 7.** Relative number abundance of all particles found at the three sites as a function of their size.

As can be seen from Figure 7, the percentage trends of the groups of particles found in the A and B sites are similar. The B site shows in the range 0.4–1 µm a larger number of particles (about 54%) than in the other ranges, which gradually decreases to 35% and 11% in the ranges 1–2.5 µm and 2.5–10 µm, respectively. The trend of the A site differs from the previous one due to the sharp decrease between the first two size ranges. In fact, the percentage frequency, with a value of 72% in the range 0.4–1 µm, is lowered to 19%

in the range 1–2.5 μm and to 9% in the range 0.4–1 μm. This means that there is a greater presence of submicrometric particles than coarse ones. A different trend is observed for the C site, which differs from the previous ones for a higher percentage of particles in the range 1–2.5 μm (48%). In the other two size ranges, i.e., 0.4–1 μm and 2.5–10 μm, the particles identified with the SEM are 37% and 15%, respectively. In order to understand which groups of particles are responsible for the differences highlighted in Figure 7, it is possible to represent in Figure 8 their relative numerical abundance in the fractions of PM1, PM2.5-1, and PM10-2.5 for the three investigated sites.

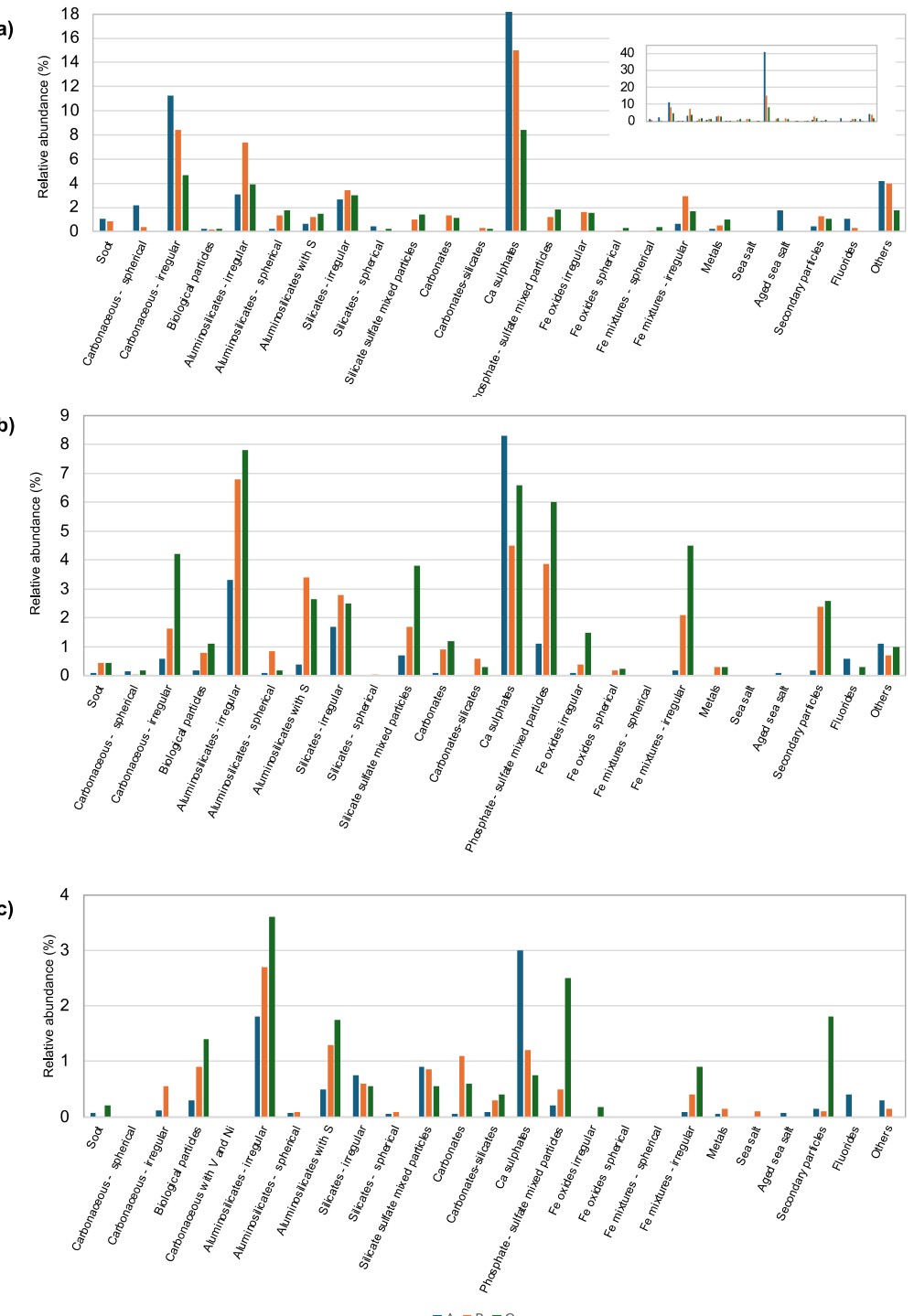

**Figure 8.** Relative numerical abundance of the groups of particles in the fractions of (**a**) PM1, (**b**) PM2.5-1, and (**c**) PM10-2.5 for the three investigated sites.

Comparing Figure 8a–c, it emerges that, in the PM1 fraction, the most relevant difference among the sites is the content of calcium sulfates. In fact, the A site is characterized by a high number of calcium sulfate particles, which represent 41% of the total PM1 particles in A (see insert in Figure 8a). This percentage drops to around 8% in the fractions of PM2.5-1 and to 3% in PM10-2.5, remaining the largest group of particles in A. The large number of calcium sulfates can be explained by the presence of construction activities near the site during the sampling period. This suggests that these particles derive essentially from the crumbling of chalky rocks or from the construction of buildings with such materials, but also from the reactions between particles rich in Ca and compounds containing S present in the atmosphere. In the PM2.5-1 fraction, the C site prevails over the other two sites due to the presence of irregular aluminosilicates, silicate sulfate mixed particles, phosphate–sulfate mixed particles, irregular Fe particles, and secondary particles. All these particles, together with the aluminosilicates with sulfur, continue to be dominant at the C site and in the PM10-2.5 fraction. It is interesting to observe that 5-day back trajectories (Figure 2) show that the mass of air belonged to NE Europe. In the literature, it is known that the enrichment of sulfate occurs with winds from this direction [34].

### 3.2. Particle Size Distributions

The numerical size distribution is shown in Figure 9 and refers to the most numerous groups of particles at each site for which it was possible to carry out a statistical analysis. The groups considered are irregular aluminosilicates, irregular silicates, aluminosilicates with sulfur, iron mixtures, phosphate–sulfate mixed particles, irregular carbonaceous particles, and calcium sulfates.

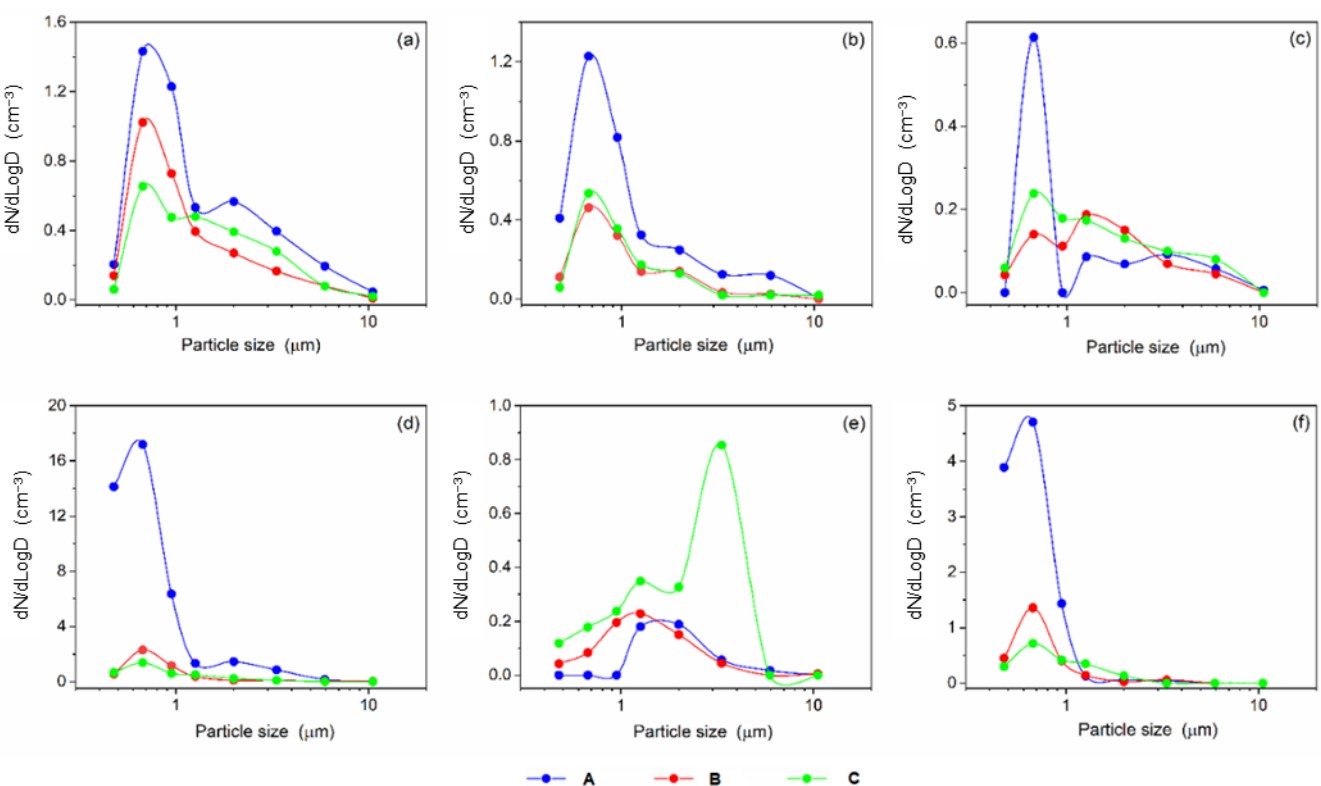

**Figure 9.** Numerical size distribution of (**a**) irregular aluminosilicates, (**b**) irregular silicates, (**c**) aluminosilicates with sulfur, (**d**) calcium sulfates, (**e**) phosphate–sulfate mixed particles, and (**f**) irregular carbonaceous particles at the A site (blue circles and line), B site (red circles and line), and C site (green circles and line).

The aerosol size distribution is quite variable in an urban area. At the three sites, the irregular aluminosilicates, irregular silicates, irregular carbonaceous particles, and calcium

sulfates have a similar numerical distribution, which is dominated by particles with diameters in the 0.6–0.8 μm size range. The highest concentration of these particles is found at the A site, which is probably closer to their sources, and at the other two sites, the concentration decreases with increasing distance from the sources [35]. Probably these are primary particles and secondary material condensed on them as they are transported through the atmosphere. Furthermore, irregular aluminosilicates, silicates, and aluminosilicates with sulfur are particles that also have a smaller concentration in the coarse fraction, showing a widened peak around 2 μm. The numerical concentration of the phosphate–sulfate mixed particles is negligible at the A and B sites, while at the C site, it follows a bimodal trend with a small peak around 1.5 μm and another more pronounced in the coarse fraction around 3.5 μm.

### 3.3. Source Apportionment

The particle groups identified in this study were assigned to the different emission sources. Based on the chemical-physical properties of the single particle, source apportionment was carried out. In fact, the chemical composition of the particles, together with their morphology, was useful to discriminate the anthropogenic component, which originates from high-temperature combustion processes, at the natural source. The source apportionment analyses were performed for the three PM fractions, i.e., PM1, PM2.5-1, and PM10-2.5, and for each sampling site. The identified sources were the following: combustion, industry, soil, SIA (secondary inorganic aerosol), sea salt, soot, carbonates, calcium sulfates, biological particles, and others. The first five sources (i.e., combustion, industry, soil, SIA, and sea salt) include multiple particle groups, while the remaining sources refer to one single group of particles and are labeled according to their chemistry and morphology. The combustion source includes irregular and spherical carbonaceous particles, metals, and spherical particles of iron oxides and mixtures. The industry includes spherical aluminosilicates and silicates. The soil includes irregular particles of aluminosilicates and silicates, aluminosilicates with sulfur, silicate–sulfate mixed particles, particles of carbonate-silicates, irregular particles of iron oxides and mixtures, fluorides, and phosphate–sulfate mixed particles. The SIA source includes only a group of secondary particles. The sea salt source includes particles of sea salt and aged sea salt. Carbonates and calcium sulfates are not included in the previous sources because they can have both natural and anthropogenic origins [13].

Figure 10 shows the mass concentration of the various sources to understand their distribution at the three sites investigated as a function of the different dimensional fractions considered.

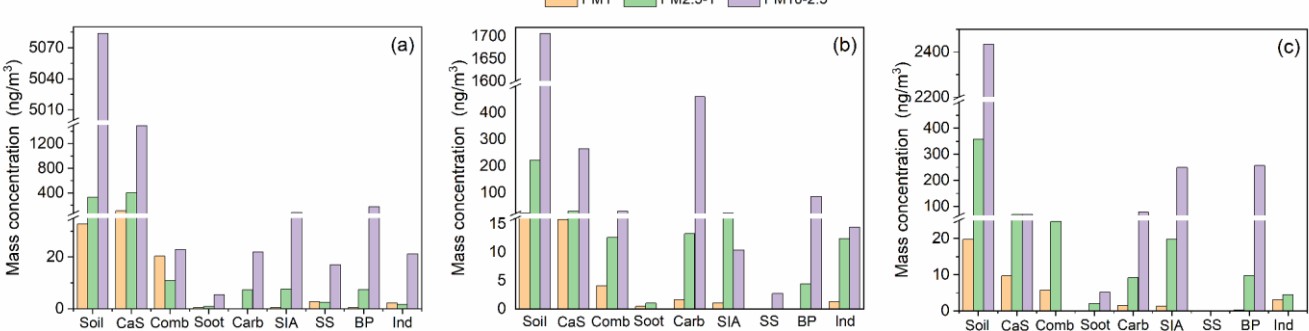

**Figure 10.** Mass concentration in the three size ranges of the soil, calcium sulfates (CaS), combustion (Comb), soot, carbonates (Carb), SIA, sea salt (SS), biological particles (BP), and industry (Ind) sources at the (**a**) A site, (**b**) B site, and (**c**) C site.

It can be observed that soil, carbonates, and biological particles are present in all the sites and in the three fractions with greater dominance in PM10-2.5. This evidence is characteristic of these sources, which emit mainly coarse particles. In the three sites,

although the average wind speed is comparable during the sampling hours, a very variable soil mass concentration is observed, with the greatest contribution at the A site. This is probably due to the different geographical positions of the sites, as the A site is located in a peripheral area of the city, surrounded by uncultivated land. For this reason, the source soil is probably more affected by the re-suspension of terrigenous material. Furthermore, the carbonate component predominates at the B site, probably affected by anthropogenic activities such as the transport of loose materials and construction works in the areas close to the site. Calcium sulfates show a concentration distribution similar to that of the previous sources, in which the presence of a relevant crustal contribution is observed. It is known from the literature that they can have both natural and anthropogenic origins and can be either a primary source, characterized by coarse-sized particles, or a secondary source with a prevalence of fine-sized particles. With single-particle analysis, it is difficult to distinguish between primary and secondary origins. The back trajectory analysis can be useful to obtain information on the transport of air masses in the Apulia region during the sampling days. Generally, when the air masses originate from Northern and Eastern Europe, they contain high levels of $SO_2$, released into the atmosphere by industries using sulfur fuel. $SO_2$ can be transported for long distances and oxidized to stable sulfate [36–38].

The combustion and industry sources are present in all size fractions, with a higher concentration in the coarse fraction at the A and B sites; at the C site, the contribution of these sources is observed only in the fine and ultra-fine fractions, and the combustion source has higher concentrations than the industry source. These are made up of different groups of particles whose anthropogenic origin is difficult to discriminate with the SEM technique alone. The SIA source is present in all three size fractions, and its mass concentration is higher in PM2.5-1. It has to be pointed out that in this study, this source is, from one side, underestimated because some of the secondary compounds are included in other sources (i.e., aluminosilicates with sulfur, etc., where sulfates are mixed with other compounds that are markers of specific sources); and on the other side, in the vacuum condition of the sample chamber of SEM and during the chemical analysis with X-rays, the volatile part of SIA could be lost. In this work, probably, also the sea salt source is underestimated as it is present in the three fractions only at the A site with a total concentration of 22.2 ng/m$^3$; at the B site it is 2.73 ng/m$^3$; and it is not observed at the C site. This is probably due to the depletion of Cl as a result of the high temperatures that occurred during the sampling period and the vacuum conditions during SEM analysis. At each site, the soot has a total mass concentration of no more than 7 ng/m$^3$, and this is indicative of sites characterized by a low impact of vehicular traffic emissions.

Figure 11 shows the percentage contribution of the different sources to PM (i.e., PM1, PM2.5-1, and PM10-2.5) for each sampling site.

At the A site, the main sources of PM1 are calcium sulfates, soil, and combustion. The percentage contributions of calcium sulfates and combustion decrease as one passes from the fine to the coarse fraction, and the resulting percentage difference is gained from the soil source. In the PM1 fraction of the B site, compared to the A site, in addition to a lower contribution from the calcium sulfate and combustion sources, the presence of industry, SIA, and carbonate sources is also observed. In PM10-2.5, carbonates increase their percentage, reaching 17.4%, resulting in the second dominant source after soil. In the fractions of PM1 and PM2.5-1, in terms of sources, the C site has similarities with the B site. In fact, in the PM1 fraction, the same sources of the B site are present (i.e., soil, calcium sulfates, combustion, carbonates, SIA, and industry), with a smaller percentage. In the PM2.5-1 fraction, there is an increase in the contribution of the soil, the SIA, and the biological particles, and consequently, a reduction of the other sources. In the PM10-2.5 fraction, in addition to the soil, which is the dominant source, the sources that contribute the most are the SIA and biological particles.

The contribution of the different sources to PM10 is shown in Figure 12 separately for the A, B, and C sites.

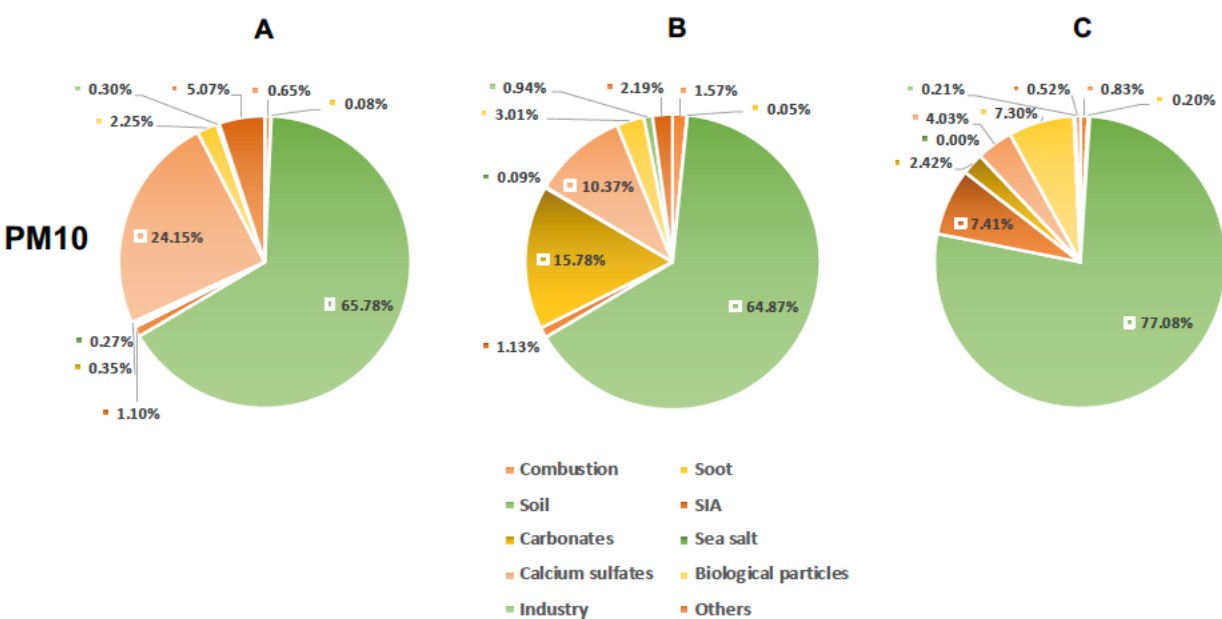

**Figure 11.** Average source contributions to PM1, PM2.5-1, and PM10-2.5 at the three measurement sites: (**A–C**).

**Figure 12.** Average source contributions to PM10 at the three measurement sites: (**A–C**).

It can be observed that, at the three sites, the soil makes the largest contribution to PM10, with 65.8% at the A site, about 65% at the B site, and 77% at the C site. Leaving aside the soil, the sources that contribute most to PM10 are calcium sulfates at the A site, carbonates and calcium sulfates at the B site, and SIA and biological particles at the C site. These sources are probably linked to the meteorological conditions present on the sampling day and to the characteristics of the site. In this study, soil is an overestimated source as it takes into account all the particles containing, in addition to the typical elements of the crustal source (i.e., Al, Si, Ca, Fe, Mg, and Na), other secondary elements such as sulfur. In fact, the particles included in this source can act as catalysts for the oxidation reaction of sulfur in the atmosphere and, therefore, be covered with secondary sulfate. Consequently, the SIA source, present at 1.10%, 1.13%, and 7.41% at the A, B, and C sites, respectively, is underestimated because some of the secondary particles are included in the soil (as already mentioned) and in the calcium sulfates. In addition, the high contribution to the SIA at the C site may be attributable to the air masses coming from the industrialized countries of Northeastern Europe, as can be seen from the analysis of the back trajectories (Figure 2) relating to the sampling day. The sea salt source has the highest contribution at the A site with 0.27%, and only aged sea salt particles are included in this category. In fact, the back trajectories of 4 June 2015, for the three altitude levels considered attest to the transit of air masses on the Adriatic Sea. At the three sites, the abundance of sea salt is variable, probably depending on the specific meteorological conditions, and therefore the number of samples investigated in this study is too low to consider a representative percentage for the sea salt source. The soot, a marker of motor vehicle traffic, also shows a very low percentage as the measurement sites are in peripheral areas of the city not affected by high-traffic roads. The industry source, which takes into account the contribution of the coal-fired power plant located about 27 km from the city, influences each sampling site by less than 1%. If we take into account that local meteorological conditions such as wind direction and speed, atmospheric stability, and rainfall can influence the transport and dispersion of pollutants, the contribution of 1% of industry to PM10 can be considered in good agreement with that calculated for the same source in Cesari et al. [9] obtained by subtracting the "crustal" contribution, identified by the Calpuff dispersion model, from the "crustal + power plant" contribution in the PMF profile (positive matrix factorization). From the analyses carried out in this study, it can be seen that the anthropogenic sources that can be held responsible for the adverse health effects impact the investigated areas with less than 3% of the total PM10. The same evidence has been reported in a parallel study on the air quality of the city of Lecce, carried out by the Regional Agency for the Prevention and Protection of the Environment of Apulia (ARPA-Puglia). Later, it was highlighted that no exceeding of the legal limits was recorded (Legislative Decree 155/10) for the various types of pollutants they monitor, including PM10 and PM2.5 [38].

## 4. Conclusions

In the southern Italian region of Apulia, in the town of Lecce, PM10 was collected between May and June 2015 at three separate locations. Based on morphology and chemical content, single particle analysis has been used in this work to categorize the various groupings of particles that make up the atmospheric particulate. To identify the main sources of aerosol at the sites, source apportionment was also performed.

The relative numerical abundance of the groups of particles reveals that the A site is characterized by calcium sulfates present in all three investigated size fractions, with a higher quantity in PM1. The presence of irregular aluminosilicates, silicate sulfate mixed particles, phosphate–sulfate mixed particles, irregular particles of Fe, secondary particles, and aluminosilicates with sulfur makes the C site stand out in the PM2.5-1 and PM10-2.5 fractions. Due to the presence of anthropogenic secondary aerosol sources in this area, the C site is distinguished by a larger presence of sulfur-containing particles. This is likely the result of air masses being transported from Northeastern Europe. The B site has a

numerical percentage of particle groups that is intermediate to the other sites in the three size fractions.

Ten sources, including combustion, industry, soil, SIA, sea salt, soot, carbonates, calcium sulfates, biological particles, and others, were identified by the source apportionment. The soil source accounts for more than 60% of the PM10 output at the three sites. The second source identified by the source apportionment technique is calcium sulfates at the A site (24.1%); carbonates and calcium sulfates source contributions at the B site are comparable (15.8% and 10.4%, respectively); SIA and biological particles are more prevalent at the C site (7.4% and 7.3%, respectively). Low levels of anthropogenic sources, such as combustion, industry, and soot, have an impact on the three sites.

In Apulia, single particle analysis performed with SEM is a technique still little used for monitoring air quality. Since with SEM it is possible to obtain detailed information on the morphology and chemical composition of the particles that make up atmospheric particulate matter, further studies are necessary to monitor the variation of PM10 over time and thus enrich the inventory of individual particles in order to attribute the latter to specific emission sources. Furthermore, as in this manuscript and in other research in the same area, it has been found that the concentration of the pollutants is low and below the legal limits in the outdoor environment. It would be appropriate to monitor indoor air pollutants in schools and houses in order to evaluate the sources present in these environments, such as home heating, lighting, cooking fumes and cigarette smoke, cleaning products, etc. The identification of the main outdoor emission sources and the further characterization of indoor air pollutants will be useful for governments in order to adopt air pollution policies to control and mitigate the adverse effects on the world and to meet the needs of future generations.

**Supplementary Materials:** The following supporting information can be downloaded at: https://www.mdpi.com/article/10.3390/su16051978/s1, Figure S1. Relative number abundance of PM10 at the three sites (A, B and C).

**Author Contributions:** T.S.: SEM analysis, data processing, writing, reviewing, and editing of the manuscript; A.D.D.: editing; F.S.: PM10 sampling and reviewing and editing; A.G.: data interpretation, reviewing, and editing of the manuscript. The authors declare that any opinions expressed in this article are their own and do not necessarily represent the opinions of the institutions to which they are affiliated. All authors have read and agreed to the published version of the manuscript.

**Funding:** This research received no external funding.

**Institutional Review Board Statement:** Not applicable.

**Informed Consent Statement:** Not applicable.

**Data Availability Statement:** Data is contained within the article and in the Supplementary Material.

**Conflicts of Interest:** The authors declare no conflict of interest.

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
