# Peer review of "Source Apportionment of PM10 as a Tool for Environmental Sustainability in Three School Districts of Lecce (Apulia)"

_sustainability, doi:10.3390/su16051978_

Round 1

Reviewer 1 Report

Comments and Suggestions for Authors

This manuscript examines the morphological and chemical properties of PM10 in three primary schools in Lecce and provides source analysis. The study is relevant, but there are still some issues that need to be revised.

1.      Each paragraph of the Introduction is short, and it is suggested that the content be revised and consolidated into 4-5 paragraphs.

2.      Three primary schools located at different traffic intensities were selected for sampling in this manuscript, how many primary schools are there in this area? Did the selection of these three schools consider the impact of surrounding industries on air quality?

3.      In the Introduction to this manuscript, the dangers of air pollution for young children are noted, but are not discussed in the manuscript, so please add them to enrich the depth of the manuscript.

4.      The Conclusion states that this manuscript is consistent with the conclusions of De Donno et al. [16], and it is suggested that this section be moved to the Results and Discussion section and that a more in-depth exploration be added.

5.      In addition to summarizing the content of the existing manuscript, the Conclusions should point out the shortcomings of the existing research and provide an outlook on the next research objectives.

6.      The formatting of references needs to be checked; there are inconsistencies in formatting and inconsistent fonts.

Reviewer 2 Report

Comments and Suggestions for Authors

I think the manuscript needs major revision before it can be considered for publication.

Reviewer 3 Report

Comments and Suggestions for Authors

Dear Author Team

overall article is very good, however there is the need specify aim and objectives, separate tasks

Figures should be of good resolution

some additional references (recommended) on analytical field chemistry might be highly recommended, eg., Field portable X-ray... 2016 J of Env Analytical Chemistry, as well as other works by Krauklis et al, Rudovica et al

Reviewer 4 Report

Comments and Suggestions for Authors

The paper presented a very actual topic. It is presented very well. I have only one comment figure 8 is not very clear. May be it will be better if it is separate at three figures. 

Round 2

Reviewer 1 Report

Comments and Suggestions for Authors

Thank you for your careful revision.

Reviewer 3 Report

Comments and Suggestions for Authors

nice work!